# A Thermosensitive, Chitosan-Based Hydrogel as Delivery System for Antibacterial Liposomes to Surgical Site Infections

**DOI:** 10.3390/pharmaceutics14122841

**Published:** 2022-12-18

**Authors:** Laurine Kaul, Clara E. Grundmann, Monika Köll-Weber, Hanna Löffler, Artur Weiz, Andrew C. W. Zannettino, Katharina Richter, Regine Süss

**Affiliations:** 1Richter Lab, Department of Surgery, Basil Hetzel Institute for Translational Health Research, University of Adelaide, 37 Woodville Rd., Adelaide, SA 5011, Australia; 2Institute of Pharmaceutical Sciences, Department of Pharmaceutics, University of Freiburg, Sonnenstr. 5, 79104 Freiburg, Germany; 3Adelaide Medical School, Faculty of Health and Medical Sciences, University of Adelaide, North Terrace, Adelaide, SA 5000, Australia; 4Precision Cancer Medicine Theme, South Australian Health & Medical Research Institute, North Terrace, Adelaide, SA 5000, Australia; 5Central Adelaide Local Health Network, 1 Port Rd., Adelaide, SA 5000, Australia; 6Institute for Photonics and Advanced Sensing, North Terrace Campus, University of Adelaide, Adelaide, SA 5005, Australia

**Keywords:** diethyldithiocarbamate, copper ions, liposomes, chitosan, beta-glycerophosphate, surgical site infections, drug delivery, thermosensitive gel, biofilms

## Abstract

Prophylaxis and the treatment of surgical site infections (SSIs) with antibiotics frequently fail due to the antibiotic resistance of bacteria and the ability of bacteria to reside in biofilms (i.e., bacterial clusters in a protective matrix). Therefore, alternative antibacterial treatments are required to combat biofilm infections. The combination of diethyldithiocarbamate (DDC^−^) and copper ions (Cu^2+^) exhibited antibiofilm activity against the staphylococci species associated with SSIs; however, the formation of a water-insoluble Cu(DDC)_2_ complex limits its application to SSIs. Here, we describe the development and antibiofilm activity of an injectable gel containing a liposomal formulation of Cu(DDC)_2_ and Cu^2+^ (lipogel). Lyophilized liposomes were incorporated into a mixture of chitosan (CS) and beta-glycerophosphate (βGP), and the thermosensitive gelling properties of CS-βGP and the lipogel were determined. The liposomes remained stable after lyophilization over six months at 4–6 °C and −20 °C. The sol-gel transition of the gel and lipogel occurred between 33 and 39 °C, independently of sterilization or storage at −20 °C. CS-βGP is biocompatible and the liposomes were released over time. The lipogel prevented biofilm formation over 2 days and killed 98.7% of the methicillin-resistant *Staphylococcus aureus* and 99.9% of the *Staphylococcus epidermidis* biofilms. Therefore, the lipogel is a promising new prophylaxis and treatment strategy for local application to SSIs.

## 1. Introduction

Surgical site infections (SSIs) are amongst the most serious complications following a surgical procedure [1] and affect up to 20% of surgeries [2]. Biofilm-forming bacteria, such as *Staphylococcus aureus* (*S. aureus*) and *Staphylococcus epidermidis* (*S. epidermidis*), are the major pathogens associated with SSIs [3], and the presence of biofilms has been reported in over 80% of SSIs [4]. Biofilms are communities of bacteria embedded in a self-produced matrix, which offers protection from the immune system and antibiotics [5]. Consequently, biofilms require higher antibiotic concentrations to control bacterial growth compared to single bacteria [6]. However, the administration of high antibiotic concentrations or administration over a prolonged period can result in toxic adverse effects [7]. In addition, traditional antibiotics that play an important role in both the prevention [3] and the treatment of SSIs [8] increasingly fail to prevent or cure SSIs due to the rise in antibiotic-resistant bacteria [9]. Therefore, identifying antibacterial agents with activity against antibiotic-resistant bacteria, such as methicillin-resistant *S. aureus* (MRSA), and their biofilms is imperative.

We previously demonstrated the antibacterial activity of a combination treatment, comprising diethyldithiocarbamate (DDC^−^) and copper ions (Cu^2+^), against *S. aureus*, MRSA, and *S. epidermidis* in vitro and in vivo [10]. In addition, we showed that there was no toxicity in both in vitro cell cultures and an invertebrate in vivo model [10]. When DDC^−^ and Cu^2+^ are mixed, a water-insoluble Cu(DDC)_2_ complex (2:1 molar ratio of DDC^−^ and Cu^2+^) is instantly formed [11]. The on-site mixing of DDC^−^ and Cu^2+^ can be performed for pre-clinical studies, but it is not suitable for clinical use [12]. Therefore, multiple carriers have recently been developed to solubilize Cu(DDC)_2_, including nanocomplexes [13,14], nanocrystals [15], biohybrid nanoparticles [16], and liposomes [11,12,17,18,19]. As Cu(DDC)_2_ exhibits activity against cancer cells [20], these nanovesicles were developed and investigated as an anti-cancer treatment, but they can be used for drug delivery in various diseases, such as infections.

Wehbe et al. [11] developed liposomes containing an aqueous Cu^2+^ core, which was then loaded with DDC^−^. DDC^−^ diffuses through the lipid bilayer and is trapped within the liposomes when the insoluble Cu(DDC)_2_ complex is formed. Hartwig et al. [17] further optimized this development process and analyzed colloidal stability and drug retention during storage. As the antibacterial activity of DDC^−^ and Cu^2+^ is based on the presence of Cu(DDC)_2_ and an excess of Cu^2+^ [10], a mix of Cu^2+^-liposomes and Cu(DDC)_2_-liposomes can be used. While this liposomal dispersion might be applied to superficial surgical wounds, a local application to other surgical sites, such as an implant, would be less effective. Therefore, the present study focused on the development of an in situ-forming depot to facilitate injection at the surgical site and control the release of antibacterial agents. 

A thermally induced gelling system comprising chitosan (CS)—a naturally occurring polysaccharide—and beta-glycerophosphate (βGP) was investigated by Chenite et al. [21]. The authors showed that CS-βGP can be administered to the body via injection as it is liquid at room temperature and forms a gel in situ at body temperature [21]. Consequently, CS-βGP has been used as a drug delivery platform for nasal, ocular, vaginal, lung, and dermal delivery [22], and a broad range of treatments have been incorporated into CS-βGP, including vaccines [23], insulin [24], anticancer drugs [25,26,27], anti-inflammatory drugs [28], antibiotics [7,29,30], and cells for tissue engineering [22,31,32]. To overcome unfavorable pharmacokinetic profiles and non-ideal properties of the drugs and to prolong or delay drug release from the gel, the drugs can be loaded into liposomes prior to CS-βGP incorporation (lipogel) [25,33,34,35,36]. However, CS-βGP as a drug delivery system for SSIs was only investigated for the antibiotic vancomycin [7], and an application of the gel against biofilms was only evaluated with incorporated zinc oxide nanoparticles against *Porphyromonas gingivalis* for peri-implant infection [37]. In addition, the development of a gel for the local delivery of DDC^−^ and Cu^2+^ has not yet been proposed.

The objective of the present study was to prepare and characterize in situ-forming lipogels containing the antibacterial agents DDC^−^ and Cu^2+^, which showed antibiofilm activity against *S. aureus* and *S. epidermidis*. To improve stability and incorporation into the hydrogel, the effect of lyophilization on colloidal stability and the storage of Cu^2+^-liposomes and Cu(DDC)_2_-liposomes was evaluated. For the preparation of an in situ thermosensitive hydrogel, CS-βGP was sterilized and used as a carrier for Cu^2+^-liposomes and Cu(DDC)_2_-liposomes. Changes in the rheological behaviors of the hydrogel, including gelation temperature, gelation time, and gel strength, caused by the addition of liposomes were evaluated. The in vitro release of liposomes was estimated by diffusion assay and weight loss measurements and through the antibacterial and antibiofilm activity against MRSA and *S. epidermidis*. 

## 2. Materials and Methods

### 2.1. Bacterial Strains, Cell Cultures, Materials, and Chemicals

*S. epidermidis* ATCC 35984 and *S. aureus* ATCC 700699 (also known as MRSA Mu50) were purchased from the American Type Culture Collection (Manassas, VA, USA). Single colonies were dissolved in 0.9% NaCl, adjusted to 0.5 McFarland units, and further 1:1000 (*v*/*v*) diluted in tryptone soya broth (TSB, Thermo Fisher Scientific, Waltham, MA, USA). Bacteria were grown in an incubator at 37 °C under aerobic conditions. Tryptone soya agar (TSA) was prepared by adding 1.5% agar bacteriological (Thermo Fisher Scientific) before steam sterilization. A10 phosphate buffer and B trace solution were prepared according to Rybtke et al. [38]. Prior to steam sterilization, B trace solution was supplemented with 1.5% agar bacteriological and 0.5% Bacto^TM^ Proteose Peptone (Thermo Fisher Scientific). The final AB trace agar contained 90% B trace agar, 10% sterile filtered A10 phosphate buffer, and 0.5% glucose. Cell culture studies were conducted using control human fibroblast cells (GM00038) obtained from the Coriell Institute for Medical Research (Camden, NJ, USA). The normal human skin fibroblast cell line was cultured in Eagle’s minimum essential medium with Earle’s salts and non-essential amino acids supplemented with 15% (*v*/*v*) fetal bovine serum (Biochrom, Berlin, Germany) and 2.2 g/L sodium bicarbonate and cultivated in cell culture flasks in an incubator at 37 °C with 5% CO_2_. The lipids for the liposome production included 1,2-distearoyl-sn-glycero-3-phosphocholine (DSPC) and 1,2-distearoyl-sn-glycero-3-phosphoethanolamine-N-[methoxy(polyethylene glycerol)-2000] (DSPE-mPEG_2000_), which were generously donated by Lipoid GmbH (Ludwigshafen, Germany), and the cholesterol (Chol) was purchased from Sigma-Aldrich (Steinheim, Germany). Unless stated otherwise, all the chemicals, materials, media, and supplements were purchased from Sigma-Aldrich.

### 2.2. Preparation of Liposomes

Cu^2+^-liposomes and Cu(DDC)_2_-liposomes composed of DSPC:Chol:DSPE-mPEG_2000_ (50:45:5, molar ratio) were prepared by the thin-film hydration method described by Hartwig et al. [17]. Briefly, the lipids were dissolved in chloroform and evaporated to dryness by rotation under reduced pressure at 65 °C using a rotary evaporator (Vacuubrand, Wertheim and VWR, Darmstadt, Germany). The film was then hydrated in a 150 mM CuSO_4_ aqueous solution. The liposomes (40 mM total lipid) were extruded at 65 °C through a 0.08 µm polycarbonate membrane (GE Healthcare Life Science, Marlborough, MA, USA) by a 1 mL Liposofast extruder (Avestin, Ottawa, ON, Canada). Unentrapped Cu^2+^ was removed by passage over a Sephadex^TM^ G-50 Fine (GE Healthcare Life Science) column with an EDTA-containing sucrose buffer (300 mM sucrose, 20 mM HEPES, 30 mM EDTA, pH 7.4). Cu^2+^-liposomes were collected following a buffer exchange to an EDTA-free sucrose buffer (SH: 300 mM sucrose, 20 mM HEPES, pH 7.4) by 3 centrifugation steps (3000× *g*, ambient temperature, 1.5 h per step), using Vivaspin^®^ Turbo 4 filtration units (100 kDa MWCO; Sartorius AG, Göttingen, Germany).

Cu(DDC)_2_-liposomes were prepared by incubation of Cu^2+^-liposomes with 70 mM DDC^−^ in water at 25 °C and mixed at 300 rpm for 10 min using a Thermomixer comfort (Eppendorf, Hamburg, Germany), allowing the DDC^−^ to pass the liposomal membrane and to complex with the entrapped Cu^2+^. Following a filtration step with a 0.45 µm cellulose acetate filter (VWR International, Radnor, PA, USA) to remove non-encapsulated Cu(DDC)_2_, excess DDC^−^ was removed during 3 centrifugation steps (3000× *g*, ambient temperature, 45 min per step), using Vivaspin^®^ Turbo 4 filtration units (100 kDa MWCO). Before use, the Cu^2+^-liposomes and Cu(DDC)_2_-liposomes were sterile filtered with a 0.2 µm cellulose acetate filter (VWR International) and characterized according to Hartwig et al. [17].

### 2.3. Liposome Characterization

#### 2.3.1. Size and Polydispersity Index

The size—expressed as hydrodynamic diameter (d_h_)—and the polydispersity index (PDI) were measured via dynamic light scattering (DLS, ZetaPals, Brookhaven Instruments Corporation, Holtsville, NY, USA). A 1:200 (*v*/*v*) dilution of Cu^2+^-liposomes and a 1.5:1000 (*v*/*v*) dilution of Cu(DDC)_2_-liposomes in SH buffer (viscosity: 1.213 mPa·s; refractive index: 1.345) were used for size and polydispersity analyses. 

#### 2.3.2. Quantification of Encapsulated Cu^2+^

The encapsulated Cu^2+^ of the Cu^2+^-liposomes and Cu(DDC)_2_-liposomes was quantitated via spectrophotometry. The absorbance of Cu^2+^ as the Cu(DDC)_2_ complex was measured at λ = 435 nm with a GENESYS 10S UV-Vis spectrophotometer (Thermo Fisher Scientific). The Cu^2+^-liposomes or Cu(DDC)_2_-liposomes were 1:10 (*v*/*v*) mixed with methanol (HPLC grade, Carl Roth, Karlsruhe, Germany), and 1 mL of the mix was measured in a 1.5 mL semi-micro polymethyl methacrylate cuvette (Brand GmbH + Co, Wertheim, Germany). Methanol disrupts the liposomal membrane and solubilizes the Cu(DDC)_2_ complex. For quantitation of the Cu^2+^-liposomes, methanol was supplemented with an excess of DDC^−^ (70 µM) to complex all the Cu^2+^ to Cu(DDC)_2_. The measured Cu(DDC)_2_ absorbance was quantitated by the linear least square regression analysis of a calibration curve, based on standard aqueous Cu^2+^ solutions (0.15–1 mM) in 70 µM DDC^−^ in methanol. All the liposomal concentrations are expressed as Cu^2+^ concentrations. The Cu(DDC)_2_-liposomes + Cu^2+^-liposomes were mixed in a 1:6.2 molar ratio of Cu(DDC)_2_-liposomes to Cu^2+^-liposomes. 

### 2.4. Lyophilization of Liposomes

The lyophilization cycle was designed based on the glass transition of the maximally freeze-concentrated amorphous phase (T_g_’) of the SH buffer. The lyophilization was conducted using the Alpha 2-4 freeze dryer (Martin Christ GmbH, Osterode am Harz, Germany) with the settings described in Table 1. Prior to lyophilization, the Cu^2+^-liposomes, Cu(DDC)_2_-liposomes, or a mix of both were diluted to a Cu^2+^ concentration of 600 µM in SH buffer, placed in glass vials, and frozen at −80 °C for 12 h. The primary drying temperature of −45 °C was selected as approximately 10 °C below the T_g_’ of the SH buffer [39]. Following the end of the secondary drying, the glass vials were sealed and stored at −20 °C.

To investigate the potential effects of the lyophilization process on the Cu^2+^-liposome and Cu(DDC)_2_-liposome stability, the d_h_ and PDI were determined following the rehydration of the lyophilizate with ultrapure water (Milli-Q system, Merck, Darmstadt, Germany; Figure 1a). To investigate the percentage of retained Cu^2+^ or Cu(DDC)_2_ within the liposomes, the Cu^2+^ concentrations were measured using UV-Vis, as described in Section 2.3.2. Following rehydration, the total Cu^2+^ concentration (encapsulated and non-encapsulated, C_[total]_) was measured. The intact Cu^2+^-liposomes were then separated from the leaked non-liposomal Cu^2+^ by centrifugation (3000× *g*, room temperature, 10 min) using Vivaspin^®^ Turbo 4 filtration units, and the intact Cu(DDC)_2_-liposomes were separated from the non-liposomal Cu(DDC)_2_ by surface area filtration (0.2 µm cellulose acetate filter). Following the filtration step, the Cu^2+^ concentrations of separated non-liposomal Cu^2+^ (C_[Cu2+]_) and liposomal Cu(DDC)_2_-liposomes (C_[Cu(DDC)2-liposomes]_) were measured. The percentage of retained encapsulated Cu^2+^ in the Cu^2+^-liposomes was calculated according to Equation (1), and the percentage of retained encapsulated Cu(DDC)_2_ in the Cu(DDC)_2_-liposomes was calculated according to Equation (2).
(1)$${\%\;\mathrm{Retained}\;\mathrm{Cu}}^{2+}\;\mathrm{within}\;\mathrm{liposome}=\frac{\left(C_{\left[\mathrm{total}\right]}-{\;C}_{\left[\mathrm{Cu}2+\right]}\right)}{C_{\left[\mathrm{total}\right]}}\times 100$$
(2)$$\%\;\mathrm{Retained}\;\mathrm{Cu}{\left(\mathrm{DDC}\right)}_{2}\;\mathrm{within}\;\mathrm{liposome}=\frac{C_{\left[\mathrm{Cu}\left(\mathrm{DDC}\right)2-\mathrm{liposome}\right]}}{C_{\left[\mathrm{total}\right]}}\times 100$$

### 2.5. Stability of Lyophilized Liposomes

For storage stability studies, aliquots of lyophilized Cu^2+^-liposomes and Cu(DDC)_2_-liposomes were kept at either 4–6 °C or −20 °C for up to 168 days (approximately 6 months). At the indicated time points, lyophilized liposomes were resuspended with the amount of water removed during the lyophilization process, characterized by DLS, and the percentage of retained Cu^2+^ or Cu(DDC)_2_ within the liposomes was determined according to Section 2.4.

### 2.6. Preparation of Hydrogel

Chitosan (CS; 95% deacetylation; molecular weight 100–250 kDa; Heppe Medical Chitosan GmbH, Halle, Germany) was dissolved in 0.15 M acetic acid solution (2% *w*/*v*) under stirring and stored at 4 °C. To prepare the gel, ice-cold βGP (13% *w/v*) was added dropwise to the CS solution up to a final molar ratio of 1:4.88 of CS to βGP and stirred in an ice bath for 15 min. The CS-βGP mix with a pH of 7.15 ± 0.17 (pH meter CG 843 P, Schott, Mainz, Germany) was either directly aliquoted and stored at −20 °C or, following a 10 min rest period in the ice bath without stirring, directly used for experiments. Following storage at −20 °C, the CS-βGP mix was thawed at ambient temperature and directly used for experiments. For the use of the CS-βGP mix under aseptic conditions, the CS solution was exposed to ultraviolet light for 20 min, and the βGP was sterile filtered through a 0.2 µm cellulose acetate membrane (sterile CS-βGP). To incorporate the liposomes, the lyophilized liposomes were resuspended in thawed or freshly prepared CS-βGP.

### 2.7. Rheological Measurements

The rheological tests were conducted with a Kinexus Lab+ rotational rheometer (Malvern Instruments, Malvern, UK), using a cone/plate geometry (CP1/40; PLS40) with a gap of 23 µm. The storage modulus (G’) and loss modulus (G’’) of the samples were measured within the linear viscoelastic range at a constant strain amplitude of 5% and a constant frequency of 0.5 Hz. The CS-βGP mixture (600 µL) was placed on the sample holder at 15 °C and sealed with a viscous paraffin solution to prevent sample evaporation during the measurement. The temperature sweep was performed from 15 to 45 °C with a heating rate of 2 °C/min. The time sweep was performed over 500 s and simulated an injection into the human body. The temperature was increased at a maximum heating rate from 15 to 37 °C for over 75 s; then, the temperature was held at 37 °C for the remaining time. The sol-gel transition temperature and time were defined as the intersection of the G’ and G’’ curves. Considering that the strength of hydrogels can be evaluated in terms of the behavior of G’ and G’’ at low frequencies [40], a frequency sweep over the range of 0.1–10 Hz was performed following the time sweep at 37 °C. The effect of sterilization, storage at −20 °C, and the incorporation of liposomes was investigated. The liposomes were added at Cu^2+^ concentrations of 128 µM.

### 2.8. Cytotoxicity of Gel

The human dermal fibroblast cells were seeded in the wells of the appropriate plate and incubated for 24 h. Following media renewal, the fibroblasts were exposed to either gel or supernatant over 24 h, as specified below. The fibroblast viability was assessed with the CellTiter-Glo^®^ Luminescent Viability Assay (Promega Corporation, Fitchburg, WI, USA) according to the manufacturer’s instructions, and luminescence was measured on a *FLx800*^TM^ Multi-Detection Microplate Reader (BioTek Instruments, Winooski, VT, USA). The percentage of fibroblast viability was calculated using Equation (3) with the luminescence intensity of the treated and untreated fibroblasts, represented by I_treatment_ and I_untreated_, respectively, and I_blank_, representing the background luminescence of the cell medium.
(3)$$\%\;\mathrm{Fibroblast}\;\mathrm{viability}=\left(\frac{I_{\mathrm{treatment}}-I_{\mathrm{blank}}}{I_{\mathrm{untreated}}-I_{\mathrm{blank}}}\right)\times 100$$

The cutoff for designating the treatment as non-toxic was determined according to ISO norm 10993-5:2009(E) with the direct contact method as viabilities exceeding 70%.

#### 2.8.1. CS-βGP Gel Covering Fibroblast Cells

In a 12-well plate, 6 × 10^5^ cells in 1.5 mL culture medium were seeded in each well and incubated for 24 h to allow attachment. Then, the fibroblast cells were covered with 250 or 500 µL sterile CS-βGP gel in the presence of 1 mL media for 24 h. 

#### 2.8.2. Fibroblast Cells Exposed to Released Components of CS-βGP Gel

In a black 96-well plate, 5 × 10^4^ cells in 200 µL culture medium were seeded in each well and incubated for 24 h. The cells were treated with a mix of 200 µL media and 100 µL of 0.9% NaCl, previously incubated with CS-βGP for 0, 6, 24, or 72 h, as described in Section 2.9.

### 2.9. Effect of Released Liposomes from CS-βGP Gel on Fibroblast Viability

The release of Cu^2+^-liposomes and Cu(DDC)_2_-liposomes from the respective lipogels was assessed by investigating the effect of the released liposomes on human dermal fibroblast cell viability. In a 96-well plate, lyophilized liposomes equivalent to 770 µM Cu^2+^-liposomes or 14 µM Cu(DDC)_2_-liposomes (based on Cu^2+^ concentrations) were incorporated into 100 µL of thawed CS-βGP mix. The solution containing liposomes and CS-βGP was heated to 37 °C for 5 min using the Thermomixer comfort to enable gel formation, then covered with 100 µL release media (0.9% NaCl). Following incubation on an orbital shaker at 37 °C and 500 rpm for 0, 4, 6, 24, and 72 h for the Cu^2+^-liposomes incorporated in the CS-βGP gel (Cu^2+^-lipogel) and 0, 0.5, 1, 2, 4, 6, 24, and 72 h for the Cu(DDC)_2_-liposomes incorporated in the CS-βGP gel (Cu(DDC)_2_-lipogel), 100 µL of release media was transferred onto the fibroblast cells (5 × 10^4^ in 200 µL culture medium, incubated at 37 °C in 5% CO_2_ for 24 h) in a black 96-well plate. The fibroblast cells were exposed to the release media for 24 h at 37 °C, 5% CO_2_. The controls were incubated for 0.5, 24, and 72 h and included (i) CS-βGP gel (see Section 2.8.2; 0% release); (ii) Cu^2+^-liposomes or Cu(DDC)_2_-liposomes in release media (100% liposomal release); and (iii) unencapsulated Cu^2+^ or Cu(DDC)_2_ in release media (100% non-liposomal release). Based on the changes observed after 24 h of incubation of the Cu(DDC)_2_-liposomes in release media, an additional measurement was taken after 6 h. The fibroblast viability was determined as described in Section 2.8. 

### 2.10. Weight Loss over Time

The weight loss of the CS-βGP gel or the CS-βGP gel with incorporated Cu(DDC)_2_-liposomes + Cu^2+^-liposomes (Cu(DDC)_2_+Cu^2+^-lipogel), either freshly prepared or stored at −20 °C, was determined over 49 days. The gels were prepared by pouring 1 mL of CS-βGP solution into transwell inserts (polyester, pore size 3 µm, whose tares are known; Corning, Kaiserslautern, Germany), weighed (gel initial weight, W_i_), placed in a 6-well plate, and heated to 37 °C in an incubator. The system consisting of the gel and the transwell insert was kept at 37 °C over a timeframe of 5 weeks, and each system was weighed at given time intervals (gel weight at timepoint t, W_t_). The percentage of weight loss at the timepoint t was calculated using Equation (4).
(4)$$\%\;\mathrm{Weight}\;\mathrm{loss}=\left(1-\frac{W_{t}}{W_{i}}\right)\times 100$$

### 2.11. Antibiofilm Activity of Gel

The antibiofilm activity of the Cu(DDC)_2_+Cu^2+^-lipogel was determined as described by Richter et al. [41]. Each side of the polycarbonate membranes with a 100 nm pore size (Whatman, GE Healthcare, Little Chalfont, UK) was exposed to UV light for 10 min prior to use. Up to 4 membranes were placed on a TSA plate, and each was inoculated with 1 µL of MRSA Mu50 or *S. epidermidis* ATCC 35984 suspension (equivalent to 1 × 10^5^ colony-forming units (CFU)/mL). To determine the prevention of biofilm growth, the bacteria were exposed to 200 µL Cu(DDC)_2_+Cu^2+^-lipogel 10 min after inoculation and incubated for 48 h. To determine the biofilm killing of Cu(DDC)_2_+Cu^2+^-lipogel, the bacterial suspension was first incubated for 24 h for MRSA Mu50 and 48 h for *S. epidermidis* ATCC 35984, to allow biofilm formation. The membranes were then transferred onto a 12-well plate containing 2 mL AB trace agar, and the biofilms were exposed to 500 µL Cu(DDC)_2_+Cu^2+^-lipogel for 4 days. The membranes were transferred into new wells containing fresh AB trace agar after 2 days. Finally, the bacteria were recovered from the membranes in 10 mL 0.9% NaCl by vortexing–sonication–vortexing (1-15-1 min), diluted, and plated on TSA for CFU counting. The controls included untreated bacteria and CS-βGP gel. Cu(DDC)_2_+Cu^2+^-lipogel and CS-βGP gel were applied in liquid form and formed a gel on the membrane during the incubation period at 37 °C in the incubator.

### 2.12. Statistical Analysis

All experiments were conducted at least in triplicate, and the results were statistically analyzed using GraphPad Prism version 9.4.1 for Windows (GraphPad Software, San Diego, CA, USA), and the statistical significance was set with an α = 0.05. The parametric data are represented by the mean ± standard deviation (SD), which was analyzed using one-way ANOVA with Dunnett’s or Tukey’s multiple comparison test, as described in the figure legend.

## 3. Results and Discussion

### 3.1. Cu^2+^-Liposomes and Cu(DDC)_2_-Liposomes Are Stable following Lyophilization

As PEGylated Cu(DDC)_2_-liposomes are only stable at 4–6 °C for up to 3 months [17], the Cu(DDC)_2_-liposomes and Cu^2+^-liposomes were lyophilized to increase storage stability and to facilitate incorporation into the drug delivery platforms. However, the physical structure of liposomes can alter during the lyophilization process, and this can lead to drug leakage. During the freezing process, the formation of ice crystals within the liposome or in the external aqueous phase can rupture the lipid bilayer. In addition, the aggregation or fusion of liposomes during the drying process can result in a size increase [39,42]. Therefore, the d_h_ and PDI were measured before lyophilization, after rehydration, and following filtration for both the Cu^2+^-liposomes and the Cu(DDC)_2_-liposomes. The quantification of the encapsulated drug was calculated by measuring the Cu^2+^ concentration prior to lyophilization (i.e., 100% control) and following filtration (Figure 1a). 

After rehydration, the size and PDI of the Cu^2+^-liposomes were slightly increased (d_h_: 123 nm; PDI: 0.06) compared to pre-lyophilization (d_h_: 110 nm; PDI: 0.05) but were restored following filtration (d_h_: 115 nm; PDI: 0.06; Figure 1b). Similar results were observed in the lyophilized Cu(DDC)_2_-liposomes, with a small increase after rehydration (d_h_: 183 nm; PDI: 0.15) compared to pre-lyophilization (d_h_: 174 nm; PDI: 0.14) and following filtration (d_h_: 170 nm; PDI: 0.13; Figure 1c). The pre-lyophilization size and PDI of the Cu^2+^-liposomes and Cu(DDC)_2_-liposomes are in accord with previously reported values [17]. In addition, Wehbe et al. [11] observed that the size determined using DLS was comparable to the vesicle size estimated by cryogenic electron microscopy (cryo-EM). The population of the liposomes was homogenous as the PDI of the Cu^2+^-liposomes and Cu(DDC)_2_-liposomes was below 0.2 prior to lyophilization and following rehydration [43]. Moreover, the Cu^2+^ retention was 72.3% for the Cu^2+^-liposomes, and the Cu(DDC)_2_ retention was 72.7% for the Cu(DDC)_2_-liposomes (Figure 1d). The change in d_h_ and PDI was not significantly different (*p* > 0.05), which may be associated with the use of sucrose as a lyoprotectant. Sucrose can help to maintain the integrity of liposomes during the lyophilization process by replacing the water molecules between the phospholipid groups. Consequently, the risk of ice crystal formation is minimized, and lipid distribution is maintained in the dry space to prevent packing defects [39,44,45]. However, the leakage of Cu^2+^ and Cu(DDC)_2_ indicated a possible rearrangement or the collapse of some liposomes. Wessman et al. [46] investigated the lyophilization of similar liposomes containing DSPC:Chol:DSPE-PEG_5000_ with the lyoprotectant lactose and the effect of osmotic stress on the liposomes. They showed a similar increase in size distribution after freeze-drying and subsequent rehydration but observed a large population of double or multi-lamellar liposomes and an increase in interbilayer distance using cryo-EM, which might be the result of an osmotic imbalance of the lyoprotectant [46]. While imaging of the Cu^2+^-liposomes and Cu(DDC)_2_-liposomes with cryo-EM previously showed mostly unilamellar versicles [11], imaging of the liposomes following rehydration can determine changes in the lamellarity and morphology of the liposomes caused by the lyophilization process. The addition of sucrose within the liposomes can prevent the osmotic stress on the outer bilayer and can therefore reduce drug leakage [44]. Moreover, another lyoprotectant can be used, such as other disaccharides or oligosaccharides [45]. Here, sucrose was used as a lyoprotectant as it is part of the buffer solution used for the preparation of the liposomes.

### 3.2. Lyophilized Liposomes Are Stable over 6 Months

The colloidal stability of the lyophilized Cu^2+^-liposomes and Cu(DDC)_2_-liposomes following storage at 4–6 °C (Figure 2a) or −20 °C (Figure 2b) was determined by changes in the d_h_ and PDI. The Cu^2+^-liposomes and Cu(DDC)_2_-liposomes stored at 4–6 °C and −20 °C over 6 months (168 days) showed no significant difference in d_h_ and PDI (*p* > 0.05). Additionally, the percentage of Cu^2+^ retained within the Cu^2+^-liposomes and the Cu(DDC)_2_ retained within the Cu(DDC)_2_-liposomes were not reduced after storage at 4–6 °C (Figure 3a) or −20 °C (Figure 3b) for up to 6 months, with 68.0% and 73.6% Cu^2+^ retention in the Cu^2+^-liposomes, respectively, and 67.6% and 70.1% Cu(DDC)_2_ retention in the Cu(DDC)_2_-liposomes, respectively. Therefore, the lyophilized Cu^2+^-liposomes and Cu(DDC)_2_-liposomes can be stored at 4–6 °C and −20 °C without significant changes in size and PDI and without drug leakage. 

Low storage temperatures facilitate the subsequent incorporation of the liposomes into the CS-βGP mix, which was either freshly prepared at 4 °C or stored at −20 °C. It was previously stated that the storage temperature should be at least 50 °C below the glass transition temperature [42], which is between 65 and 77 °C for sucrose [39]. Therefore, the storage temperatures of −20 °C and 4–6 °C are in line with these recommendations. 

### 3.3. The Sol-Gel Transition of CS-βGP Is Temperature Sensitive

The thermosensitive properties of CS-βGP (i.e., the transition from sol to gel) were measured at different temperatures by rheological measurements. While body temperature in a healthy human is around 37 °C, the temperature can drop during long surgical procedures but should not be lower than 36 °C [47]. Therefore, sol-gel transition temperatures of CS-βGP should not exceed 36–37 °C. While the sol-gel transition temperature and time differed between the tested CS-βGP mixes, all the rheological curves exhibited viscoelastic properties. Figure 4 displays the representative curves of a temperature sweep (Figure 4a) and a time sweep (Figure 4b) of CS-βGP stored at −20 °C. Both rheological measurements started at a temperature of 15 °C with G′ < G″, corresponding to higher viscous properties compared to elastic properties, resulting in a liquid form of the thawed CS-βGP. In Figure 4a, the temperature was increased from 15 °C to 45 °C over 15 min, resulting in a steady increase in G′, while G″ slightly decreased. At 40.0 °C, G′ > G″, indicating the sol-gel transition and consequent formation of CS-βGP gel. Following the sol-gel transition and until reaching 45 °C, the difference between G′ and G″ continuously increased. Moreover, the sol-gel transition time was determined when the temperature was increased from 15 to 37 °C in 75 s (maximum heating rate) and maintained at 37 °C for 425 s, mimicking an injection into the human body (Figure 4b). While with G′ < G″ at 15 °C, the rapid rise in temperature resulted in an increase in G’, while G’’ remained constant. After a total time of 125 s or 50 s after reaching 37 °C, G′ > G″, indicating the sol-gel transition and the formation of CS-βGP gel.

As described in Table 2, the sol-gel transition of non-sterilized CS-βGP, sterilized CS-βGP, Cu^2+^-lipogel, and Cu(DDC)_2_+Cu^2+^-lipogel was below body temperature and showed negligible differences in temperatures, ranging from 33.3 to 35.3 °C. Similarly to the results of the temperature sweep, the sol-gel transition times of sterilized CS-βGP, the Cu^2+^-lipogels, and the Cu(DDC)_2_+Cu^2+^-lipogels were equal to or below the 75 s required to reach 37 °C. In contrast, the Cu(DDC)_2_-lipogels, thawed CS-βGP, and thawed Cu(DDC)_2_+Cu^2+^-lipogels increased the sol-gel transition temperatures above body temperature, with temperatures ranging from 37.9 to 39.2 °C. However, these mixes resulted in a sol-gel transition over time when held at 37 °C. The Cu(DDC)_2_-lipogels showed the longest sol-gel transition time, with a total of 330 s and 255 s once reaching 37 °C. The sol-gel transition times of the thawed CS-βGP and thawed Cu(DDC)_2_+Cu^2+^-lipogels were 90 s and 118 s, respectively, corresponding to 15 s and 43 s after reaching 37 °C, respectively. 

The sterilization process of CS-βGP only slightly reduced the sol-gel transition temperature compared to non-sterilized CS-βGP, which could be the result of minor CS polymer degradation. It was previously determined that the sterilization of CS and βGP should be performed separately as terminal sterilization of the temperature sensitive CS-βGP is impossible with methods relying on steam or dry heat. Sterile filtration of the gel is not possible due to viscosity, and sterilization with gamma radiation causes degradation of the CS polymer chains. The most common sterilization processes involve sterile filtration of the βGP solution and steam sterilization of the CS powder in water and the dissolving of CS by adding acetic acid [48]. However, this sterilization process of CS resulted in sol-gel transition temperatures around 15 °C (Appendix A), which could be a result of direct damage to the CS polymers [22]. While heat sterilizing the CS powder prior to solubilization was suggested to be a suitable sterilization method, contradictory results were reported [49]. In contrast to the literature reporting degradation of the CS polymer under ultraviolet light [48], we successfully sterilized the CS solution using ultraviolet light. 

The differences in the sol-gel transition temperatures and times, caused by the storage conditions or the incorporation of liposomes, can be associated with changes in the mechanisms behind the formation of the thermosensitive gel. The mechanisms for the formation of the CS-βGP gel were previously described by Saravanan et al. [32]. The addition of βGP to the acidic CS solution results in the pH increasing to a physiological range. The phosphate groups of the βGP are attracted through electrostatic forces to the amino groups of CS, resulting in a protective layer of water molecules around the CS polymers. Due to this protective layer, the CS-βGP remains in solution at low temperatures. However, when temperatures are increased, the water molecules are scattered, resulting in hydrophobic interactions of the CS polymers and, consequently, in the formation of a gel [32]. A CS-βGP gel of similar composition and with comparable thermosensitive properties was previously determined to not be thermo-reversible by showing no sol-gel-sol reversibility with decreasing temperatures [50].

Adding Cu^2+^-liposomes did not affect the sol-gel transition temperature and time, however, adding Cu(DDC)_2_-liposomes increased the temperature above 37 °C. As both liposomes contain the same lipids [12,17], the difference cannot be attributed to liposome charge but can be a result of size difference. As the d_h_ of Cu(DDC)_2_-liposomes is larger compared to Cu^2+^-liposomes, liposome interference with CS cross-linking can occur [34]. The incorporation of Cu(DDC)_2_-liposomes + Cu^2+^-liposomes did not change the sol-gel transition time, which can be explained by the molar ratio of the combination, consisting of a higher amount of Cu^2+^-liposomes compared to Cu(DDC)_2_-liposomes. This could be verified by determining the sol-gel transition time of the lipogels containing different concentrations of Cu(DDC)_2_-liposomes. Furthermore, the resuspension of the lyophilized liposomes in CS-βGP could affect the size and morphology of the liposomes or lead to liposomal aggregation, which should be visually assessed by cryo-EM [51]. CS can coat liposomes by forming polymeric layers that modify the liposomal surface and lead to an increase in liposome size [52].

The CS-βGP was stored at −20 °C as storage at 4–6 °C was previously reported to result in sol-gel transition over time when using CS with a high degree of deacetylation [48]. After storage of the CS-βGP at −20 °C, the thawed preparation was liquid, the sol-gel transition temperature was above body temperature, and the sol-gel transition time at 37 °C increased compared to the freshly prepared CS-βGP. We suggest that during the freezing process, the water molecules that formed the protective layer crystalized, which reduced the space between the CS polymers by disrupting the electrostatic attraction between CS and βGP. This was previously observed in a CS solution with NaCl, which showed an increased positive charge following a freeze–thawing cycle to −20 °C and a more concentrated CS solution [53]. In addition, the reduced space between polymer chains after freezing was also reported as the base of CS cryogels [54]. Consequently, following the thawing process, the interaction between CS and βGP and the protective hydration layer need to be restored, which is further hindered by the proximity of the CS polymers. Once restored, the hydrophobic interactions of the CS polymer can lead to the gel formation with increasing temperature. 

Based on the viscoelastic properties determined using rheological measurements, the CS-βGP system is closer to a liquid than to a solid form at ambient temperature. To guarantee an effortless application at the surgical site, the formulation should be prepared (i.e., thawed and/or mixed with liposomes) and administered as a fluid at temperatures below 33 °C. Once injected at the site of action, the physiological temperature induces the formation of the Cu(DDC)_2_+Cu^2+^-lipogel within one minute to prolong drug exposure. 

### 3.4. Mechanical Strength of CS-βGP Gel

The gel strength was measured by observing G′ and G″ over a frequency sweep (Figure 5). The strong gel behavior of CS-βGP gels can be attributed to G′ >> G″ [48] and an intact gel structure to G′ and G″ remaining almost parallel to each other over the whole frequency range. The disruptions of the CS interactions and the breaking up of the gel structure result in changes in G′ and G″, such as a decrease in G’ while G″ remains constant [40,55]. In Figure 5a, a representative curve of CS-βGP gel at 37 °C shows that G′ > G″ and both properties remained parallel to each other, indicating a gel-like behavior. To determine the effect of sterilization, storage at −20 °C, and the incorporation of liposomes on the gel strength, the ratio of G′/G″ was observed with increased frequency (Figure 5b). While the G′/G″ ratio of the Cu(DDC)_2_-lipogel was reduced with increasing frequency, indicating a destabilization of the gel structure, CS-βGP, sterile CS-βGP, −20 °C stored CS-βGP, the Cu^2+^-lipogels, and the Cu(DDC)_2_+Cu^2+^-lipogels remained constant, suggesting a stable gel structure up to 10 Hz. The destabilization of the gel structure through incorporation of Cu(DDC)_2_-liposomes and the increased sol-gel transition temperature and time further suggest the interference of Cu(DDC)_2_-liposomes with the cross-linking of CS polymers. To characterize the gel and lipogel, scanning electron microscopy can be used to investigate the pore size of the CS-βGP structure and to visualize the liposomes in the gel [25,56].

### 3.5. CS-βGP Is Biocompatible

CS is a non-toxic, biodegradable, and biocompatible polymer approved by the US Food and Drug Administration (FDA). Similarly, βGP, which is found naturally in the body, has been approved by the FDA for intravenous administration for the treatment of phosphate metabolism imbalances [22]. To determine the effect of CS-βGP on human cells, the human dermal fibroblast cells were covered with CS-βGP (Figure 6a). The CS-βGP showed negligible cytotoxic effects, with over 70% fibroblast viability. A slight fibroblast viability reduction from 89.0% to 74.1% was observed with the increasing amount of CS-βGP, which can be explained by the limited access of the fibroblast cells to oxygen and nutrients in vitro [57]. To assess the effect of the components released from the CS-βGP on human cells, the CS-βGP was incubated for up to 72 h with 0.9% NaCl, before the supernatant was transferred onto the fibroblast cells and incubated for 24 h (Figure 6b). The fibroblast viability remained above 70%, indicating no toxicity of components released from the gel over 72 h. The cytotoxicity of CS-βGP was previously observed to be dependent on the deacetylation degree of CS, the solvent used to dissolve CS, and the βGP concentrations. Therefore, the biocompatibility of CS-βGP can be explained by the high deacetylation degree of CS [58] and the use of acetic acid to dissolve CS [59] and on βGP concentrations below 15% or 1.115 M [48,60,61]. 

### 3.6. Released Liposomes from CS-βGP Gel Affect Fibroblast Viability

Pharmaceutical drug release assays for gel formulation typically require big volumes of release media, and therefore, they require either very sensitive quantitative methods or high concentrations of compounds [62]. The quantification of the Cu^2+^-liposomes and Cu(DDC)_2_-liposomes was performed by measuring the Cu(DDC)_2_ complex absorbance with UV spectrophotometry. However, the antibacterial concentration of Cu(DDC)_2_ + Cu^2+^ is 128 µM [10], and the detection limit of this method is 100 µM Cu^2+^, rendering the detection of small concentrations challenging. Therefore, the release of Cu^2+^-liposomes and Cu(DDC)_2_-liposomes from the respective lipogels over time was investigated by measuring the effect of released liposomes on fibroblast viability. The lipogels were incubated with 0.9% NaCl for up to 72 h; the release medium was then transferred onto fibroblast cells and incubated for 24 h; then, the fibroblast viability was measured (Figure 7a).

When the fibroblast cells were incubated with the release media containing free Cu^2+^, the viability remained at approximately 84% over 72 h (Figure 7b), which indicated no toxicity and was similar to the effect of CS-βGP gel alone (Figure 6b). In contrast, the fibroblast cells incubated with supernatant containing the same concentration of Cu^2+^-liposomes resulted in reduced fibroblast viability, ranging from 1 to 13%, thereby permitting a distinction between the effect on the fibroblast viability of released free Cu^2+^ and of released Cu^2+^-liposomes. The difference in fibroblast viability can be associated with free Cu^2+^ interacting with components of the gel, such as CS [63] and therefore a lower Cu^2+^ concentration being transferred onto the fibroblast cells. In addition, no changes in fibroblast viability were observed over time when exposed to Cu^2+^-liposomes, suggesting the stability of Cu^2+^-liposomes and no leakage over 72 h at 37 °C. The Cu^2+^-lipogels showed no effect of Cu^2+^-liposomes in the first 6 h, with fibroblast viability remaining at approximately 78%, indicating no release of Cu^2+^-liposomes. However, following the incubation of the lipogel for 24 and 72 h, the fibroblast viability was reduced to 62% and 52%, respectively, indicating a release of Cu^2+^-liposomes from the gel. Therefore, we assume that Cu^2+^-liposomes were not released within the first 6 h and then were slowly released from the gel, reaching concentrations corresponding to approximately 50% fibroblast viability after 72 h incubation (Figure 7b). 

The same experiment was performed with the Cu(DDC)_2_-lipogels (Figure 7c). When the fibroblasts were incubated with release media containing free Cu(DDC)_2_, the viability remained between 85 and 95% over 72 h. Due to the low water solubility of Cu(DDC)_2_, the complex can sediment into the gel structure or to the surface of the well and fail to be transferred with the supernatant onto the fibroblast cells. In addition, the bioavailability of precipitated substances for cellular uptake is reduced [64]. In contrast, when the fibroblast cells were exposed to the Cu(DDC)_2_-liposomes, the viability was reduced to 2% and 5% after 0.5 and 6 h, respectively, before increasing to 15% and 30% after 24 and 72 h, respectively. This indicates that the Cu(DDC)_2_-liposomes were not stable over 72 h and that water-insoluble Cu(DDC)_2_ leaked from the liposomes. Leaked Cu(DDC)_2_ sedimented and was not transferred onto the fibroblast cells with the release media, resulting in increased fibroblast viability compared to the intact Cu(DDC)_2_-liposomes. The release media of the Cu(DDC)_2_-lipogels showed fibroblast viability below 21% within the first 6 h, suggesting an instant release of the Cu(DDC)_2_-liposomes from the gel. Similar to the Cu(DDC)_2_-liposome control, the fibroblast viability increased after 24 h and 72 h, reaching 31% and 77%, respectively, and correlating with the Cu(DDC)_2_-liposomes being unstable. Therefore, the effect on the fibroblast viability indicated a release of Cu(DDC)_2_-liposomes within the first 0.5 h and unstable Cu(DDC)_2_-liposomes after 24 h. 

The effect of the release media incubated with the lipogel on fibroblast viability can be associated with the release of Cu^2+^-liposomes and Cu(DDC)_2_-liposomes, but this is an indirect assessment of the liposomal release behavior from the lipogel. The integrity of the liposomes released from the gel should be verified by cryo-transmission electron microscopy [12]. As the release media was not replaced to investigate the effect on the fibroblast viability of the accumulated release of Cu(DDC)_2_-liposomes or Cu^2+^-liposomes, the concentration in the release media and the concentration remaining in the gel could reach equilibrium. This could potentially mask the further release of Cu^2+^-liposomes which would otherwise have occurred. This should not be the case for Cu(DDC)_2_-liposome release, as the leakage of insoluble Cu(DDC)_2_ from the liposome would create a new concentration gradient for further release of Cu(DDC)_2_-liposomes. In addition, the effects of hydrophobic Cu(DDC)_2_ being released from the liposomes on the CS–CS hydrophobic interactions and the potential of Cu^2+^ to chelate CS, thereby producing gel matrices [63] and altering the CS-βGP, were not investigated. While smaller particles are expected to diffuse faster than more voluminous particles [33], the interference of Cu(DDC)_2_-liposomes with the gel matrix, which also caused the increased sol-gel transition temperature and the reduced gel strength, could explain the burst release. In addition, liposome release cannot only be attributed to diffusion processes, but might also be a result of erosion and water escaping the gel structure [25].

### 3.7. Weight Loss of CS-βGP Gel and Cu(DDC)_2_+Cu^2+^-Lipogel over Time

Typically, an erosion assay is performed in a solution containing lysozyme to mimic the enzymes present in physiological fluids in order to assess degradability of the gel over time [65]. However, as the antibacterial assays are performed without the presence of liquid, we measured the weight loss to show a release of the liposomes under dry conditions. The percentage weight losses of the CS-βGP gels and lipogels over time are described in Figure 8a for freshly prepared gels and Figure 8b for CS-βGP stored at −20 °C. One-phase association kinetics was observed independently of the storage conditions and incorporation of liposomes. All the gels reached a plateau after 21 days, and the weight loss remained constant until day 49. In addition, 50% weight loss was reached for all the gels within a range of 2.6 to 3.9 days. The CS-βGP stored at −20 °C reached 50% weight loss in a smaller time period compared to the freshly prepared gel (−0.8 days without liposomes and −1.1 days with liposomes) and the incorporation of liposomes also slightly reduced the time until 50% weight loss compared to the gels without liposomes (−0.3 days in fresh gels and −0.5 days in −20 °C stored). This was also observed by the increased rate constants of the lipogels compared to the rate constants of the CS-βGP gels without liposomes (Table 3). While most of the weight loss can be attributed to water escaping the gel, the increased rate constant of the lipogels suggest that the liposomes are released with the water from the gel structure. This can be confirmed by measuring the Cu^2+^ concentration of the released water or by determining the antibacterial activity of the lipogel under similar conditions.

### 3.8. Antibiofilm Activity of Cu(DDC)_2_+Cu^2+^-Lipogel

The lipogels were investigated for the prevention of MRSA Mu50 and *S. epidermidis* ATCC 35984 biofilm growth (Figure 9a) and the inhibition of formed biofilms (Figure 9b). The concentrations were chosen based on the weight loss assay as the gel was not in contact with any fluid; therefore, the estimated release behavior (Section 3.6) might not correlate with the release under dry conditions. The optimal concentration was previously determined to be a 1:6.2 molar ratio of Cu(DDC)_2_ to Cu^2+^ with a total Cu^2+^ concentration of 128 µM [10]. Under the assumption that during weight loss, the liposomes are released from the gel, 30% and 50% of liposomes would be available after 2 days and 3–4 days, respectively. Therefore, we chose to investigate the lipogels containing 256 µM (128 µM after 3–4 days) and 512 µM (256 µM after 3–4 days) Cu(DDC)_2_ + Cu^2+^.

The exposure to CS-βGP and 256 µM Cu(DDC)_2_+Cu^2+^-lipogel did not prevent MRSA Mu50 biofilm formation over 2 days (log_10_ 9.3 and 9.1 CFU/membrane, respectively) compared to the untreated control (log_10_ 9.3 CFU/membrane, *p* > 0.05). In contrast, no MRSA Mu50 bacteria were detected when exposed to 512 µM Cu(DDC)_2_+Cu^2+^-lipogel. Therefore, we observed no antibacterial activity of the CS-βGP gel and prevention of biofilm growth by the lipogels in a concentration-dependent manner. The 256 µM Cu(DDC)_2_+Cu^2+^-lipogel did not prevent biofilm growth, which can be explained by the amount of liposomes released from the gel over 2 days based on the weight loss assay (Figure 8b). As only 30% of the liposomes—corresponding to approximately 80 µM—would be released after 2 days, the minimum concentration for the biofilm prevention of 128 µM Cu(DDC)_2_ + Cu^2+^ would not be reached. In contrast, the 512 µM Cu(DDC)_2_+Cu^2+^-lipogel, which would release approximately 150 µM after 2 days, reached the minimum liposome concentration needed for antibacterial activity to prevent biofilm growth. Consequently, the antibacterial activity is associated with the release of liposomes from the gel, which confirms the hypothesis that under the condition of the weight loss assay, the liposomes are released from the gel structure with water. 

When *S. epidermidis* ATCC 35984 was exposed to CS-βGP alone, no bacteria were detected after 2 days, suggesting the antibacterial activity of CS-βGP against *S. epidermidis* specifically. As *S. epidermidis* is a facultative anaerobe, the observed eradication of bacteria cannot be linked to the limited oxygen supply, which can even promote biofilm formation by enhancing the production of cell-adhesion and cell-promoting molecules [66]. CS-βGP mixes based on different concentrations, with α,βGP or with chitosan derivates previously inhibited *Porphyromonas gingivalis* and *S. aureus* growth [67,68] and postponed *Escherichia coli* growth [69]. The antibacterial activity of the different CS-βGP mixes was attributed to the antibacterial activity of the chitosan polymer [68,69,70]. For example, low molecular weight chitosan (107 kDa) at subinhibitory concentrations previously showed a significant reduction in the metabolic activity of *S. epidermidis* biofilms but not *S. aureus* biofilms [71]. In addition, Carlson et al. [72] showed the microbe-specific efficacy of chitosan coatings, based on the varying cell surface charges and differences in cell wall and membrane structure. They also showed extensively reduced *S. epidermidis* biofilm formation, compared to no significantly reduced *S. aureus* biofilm formation [72]. As eradication of *S. epidermidis* was already seen with the CS-βGP, both concentrations of the Cu(DDC)_2_+Cu^2+^-lipogel resulted in the same biofilm growth inhibition (Figure 9a).

A 512 µM Cu(DDC)_2_+Cu^2+^-lipogel concentration was chosen for the biofilm experiments as this concentration achieved the bacterial eradication of planktonic MRSA Mu50 (Figure 9a). The MRSA Mu50 and *S. epidermidis* ATCC 35984 biofilms treated with CS-βGP resulted in statistically significant log_10_ reduction compared to the untreated control, corresponding to 93.7% and 98.7% biofilm killing, respectively (Figure 9b). The treatment with 512 µM Cu(DDC)_2_+Cu^2+^-lipogel also significantly reduced the MRSA Mu50 and *S. epidermidis* biofilms compared to the untreated control, with 98.2% and 99.9% biofilm killing, respectively. While no significant difference was observed between the CS-βGP gel and the Cu(DDC)_2_+Cu^2+^-lipogel, the log_10_ of the Cu(DDC)_2_+Cu^2+^-lipogel was reduced compared to that of the CS-βGP gel.

Exposure to the Cu(DDC)_2_+Cu^2+^-lipogel successfully prevented MRSA and *S. epidermidis* biofilm formation over 2 days but did not eradicate the pre-formed biofilms. As bacteria in biofilms can be up to 1000-fold less susceptible to antimicrobial agents compared to planktonic bacteria [73], the reduced activity of the lipogel against the biofilms can be associated with the presence of a protective matrix and a stratified profile. Colony biofilms are subject to oxygen and nutrient gradients, creating layers of bacteria in different metabolic states [74]. The reduced antibiofilm activity of the lipogel can be investigated by evaluating the penetration of the released liposomes in different layers of the biofilm using fluorescently labelled liposomes, live/dead staining of bacteria, and microscopical analysis [41,75]. To further enhance the antibiofilm activity, a CS-βGP gel containing liposomes can be combined with antibiotics. DDC^−^ and Cu^2+^ previously showed synergistic effects in vitro with a range of different antibiotics [10]; therefore, the antibiofilm properties of the gel in combination with antibiotics should be examined. In addition, the colony biofilm assay provides a continuous flow of nutrients through the membrane, but the effects of host matrix components, wound simulating media, a 3D biofilm gradient, and a polymicrobial biofilm were not investigated. Therefore, the antibiofilm activity of the lipogel should be examined in an in vitro model that mimics the wound environment of SSIs [76]. The bacterial species chosen for the prevention and antibiofilm assay were MRSA and *S. epidermidis*, as they are mostly found in SSIs. However, isolated bacterial species vary depending on the surgical procedure, and staphylococci are not typically found in gastrointestinal and urological surgeries, which are mostly caused by Gram-negative and anaerobic bacteria [77,78]. As Cu(DDC)_2_ + Cu^2+^ previously showed no activity against Gram-negative bacteria [10], the application of the lipogel is restricted to the SSIs caused by staphylococci.

Consequently, the Cu(DDC)_2_+Cu^2+^-lipogel should be further evaluated for antibiofilm activity in vitro and as a preventive antibacterial therapy of surgical sites in vivo, including surgical wounds and medical devices, such as hernia meshes and joint replacements.

## 4. Conclusions

A thermosensitive CS-βGP gel with incorporated DDC^−^ and Cu^2+^ was evaluated as a treatment for surgical site infections. The findings reveal that CS-βGP is an injectable and biocompatible gel, which can be stored frozen and thawed prior to use. The lyophilization of Cu(DDC)_2_-liposomes and Cu^2+^-liposomes increased the storage stability and facilitated the incorporation into CS-βGP, without affecting its thermosensitive properties. Liposomes were released from the lipogel over time through diffusion processes and gel mass reduction due to weight loss, resulting in high in vitro antibiofilm activity against MRSA and *S. epidermidis* by preventing biofilm formation and reducing the viability of the formed biofilms. Future work will examine the in vitro antibiofilm activity of the lipogel in combination with antibiotics, and in vivo studies will determine the efficacy and safety of the lipogel. 

## 5. Patents

K.R. holds intellectual property on the Cu(DDC)_2_ + Cu^2+^ antibacterial treatment (PCT/AU2020/050661).

## Figures and Tables

**Figure 1 pharmaceutics-14-02841-f001:**
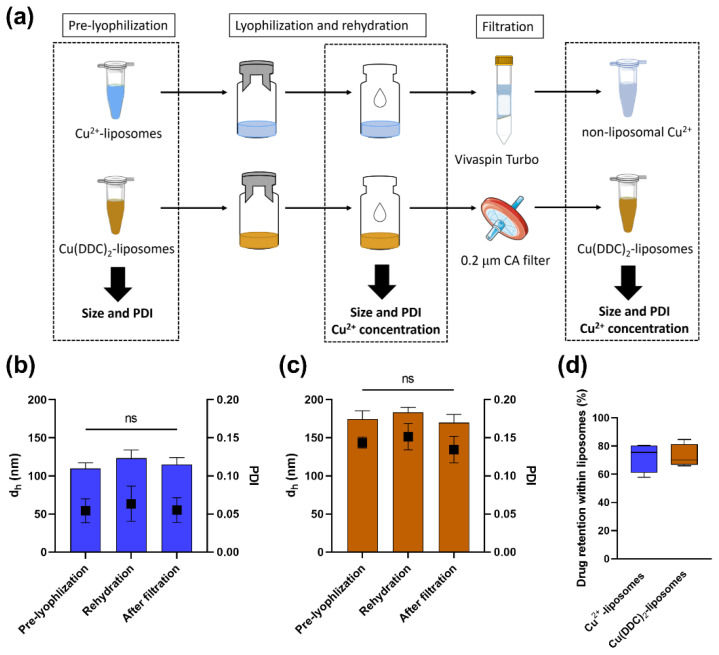
Effect of the lyophilization processes on Cu^2+^-liposomes (blue) and Cu(DDC)_2_-liposomes (brown) characteristics. (**a**) Experimental procedure of characterization steps prior and following the lyophilization process. The filtration step separated non-encapsulated Cu^2+^ and Cu(DDC)_2_. Hydrodynamic diameter (d_h_; bars) and polydispersity index (PDI; squares) were determined prior to lyophilization, after rehydration and following filtration. Total Cu^2+^ concentrations were determined after rehydration, while Cu^2+^ concentrations of separated non-encapsulated Cu^2+^ and Cu(DDC)_2_-liposomes were determined following filtration and were used to calculate drug retention (%). d_h_ and PDI of (**b**) Cu^2+^-liposomes and (**c**) Cu(DDC)_2_-liposomes prior to lyophilization, when rehydrated and following filtration. (**d**) Drug (Cu^2+^ or Cu(DDC)_2_) retention within Cu^2+^-liposomes or Cu(DDC)_2_-liposomes after lyophilization. Data are expressed as the mean ± standard deviation (n = 4; repeated measures 1-way ANOVA: ns = not significant *p* > 0.05). CA = cellulose acetate.

**Figure 2 pharmaceutics-14-02841-f002:**
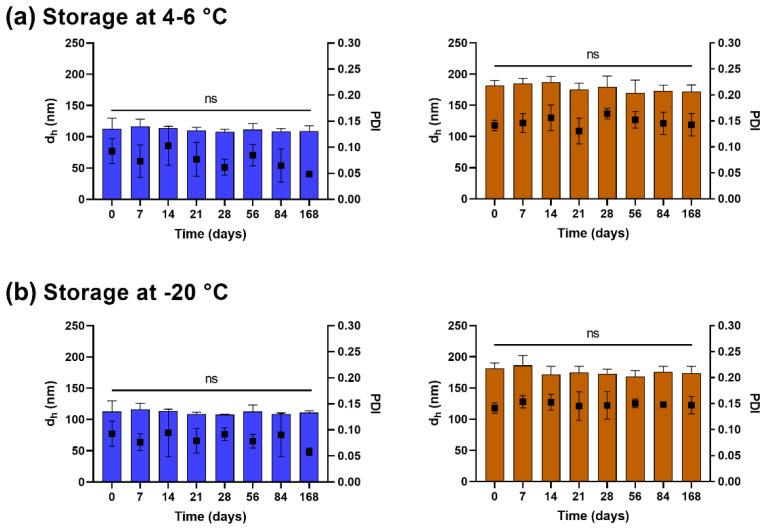
Colloidal stability analysis of rehydrated Cu^2+^-liposomes (blue) and Cu(DDC)_2_-liposomes (brown) after storage in lyophilized form at (**a**) 4–6 °C and (**b**) −20 °C. Aliquots of lyophilized Cu^2+^-liposomes and Cu(DDC)_2_-liposomes were stored for up to 168 days and resuspended, and the hydrodynamic diameter (d_h_; bars) and polydispersity index (PDI; squares) were determined via dynamic light scattering. Data are expressed as the mean ± standard deviation (n = 3; 1-way ANOVA with Dunnett’s multiple comparison test, ns = not significant *p* > 0.05).

**Figure 3 pharmaceutics-14-02841-f003:**
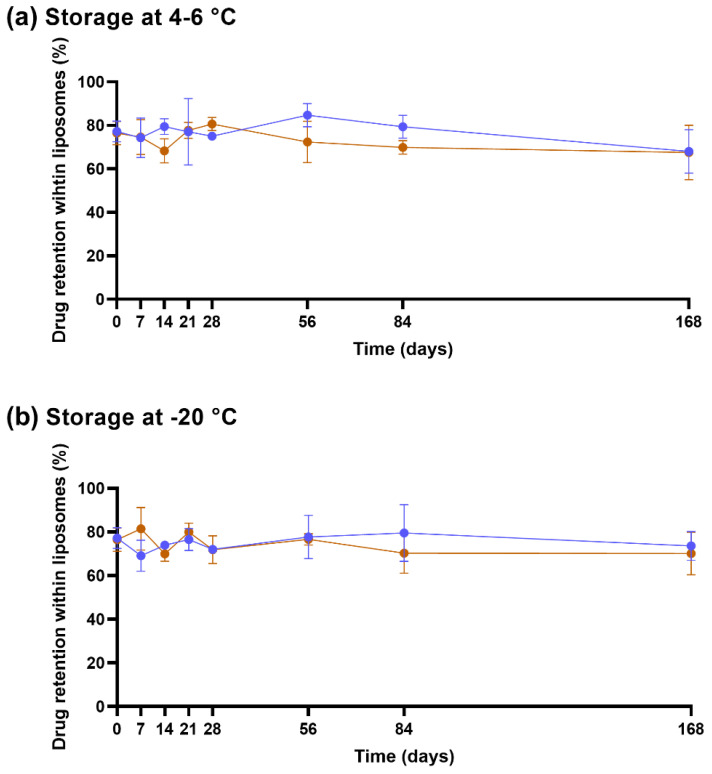
Retention of Cu^2+^ in Cu^2+^-liposomes (blue) and Cu(DDC)_2_ in Cu(DDC)_2_-liposomes (brown) following storage of the lyophilized liposomes at 4–6 °C or −20 °C. Aliquots of lyophilized Cu^2+^-liposomes and Cu(DDC)_2_-liposomes were stored at (**a**) 4–6 °C or (**b**) −20 °C for up to 168 days. Cu^2+^-liposomes and Cu(DDC)_2_-liposomes were resuspended and filtered after predetermined storage times. Cu^2+^ concentrations of Cu^2+^-liposomes and Cu(DDC)_2_-liposomes were determined before and after the filtration step. A reduction in Cu^2+^ concentration indicated Cu^2+^ or Cu(DDC)_2_ leakage from liposomes. Data are expressed as the mean ± standard deviation (n = 3).

**Figure 4 pharmaceutics-14-02841-f004:**
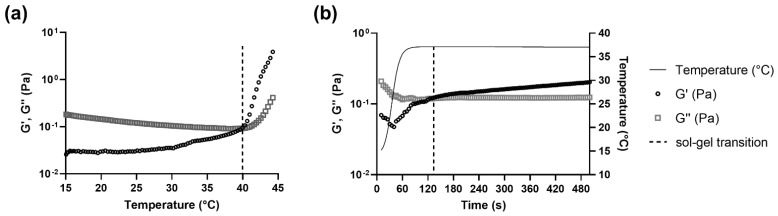
Representative rheological curves of temperature- and time-dependent sol-gel transition of sterilized chitosan with β-glycerophosphate (molar ratio 1:4.88) stored at −20 °C. Measurement of the storage modulus (G′) and loss modulus (G″) during a (**a**) temperature sweep and (**b**) over time when simulating an injection (temperature rises from 15 to 37 °C in 75 s). The dotted lines represent the temperature and/or time at the sol-gel transition.

**Figure 5 pharmaceutics-14-02841-f005:**
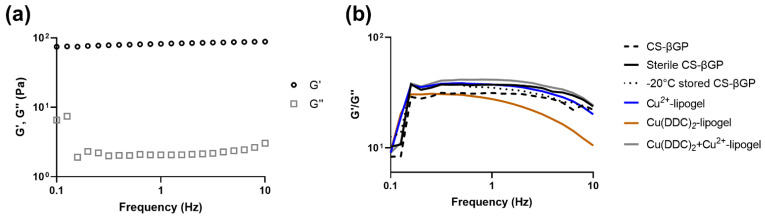
Frequency sweep (0.1–10 Hz) of chitosan with β-glycerophosphate (CS-βGP) gels and lipogels at 37 °C. (**a**) Representative curve of storage modulus (G′) and loss modulus (G″) of sterile CS-βGP gel and (**b**) ratio of G′/G″ of CS-βGP gels exposed to sterilization procedures, stored at −20 °C or with incorporated liposomes. Data are expressed as mean (n = 3).

**Figure 6 pharmaceutics-14-02841-f006:**
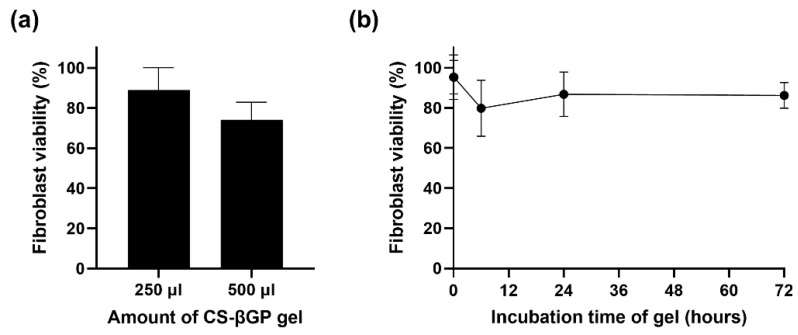
Effect of sterile chitosan with β-glycerophosphate (CS-βGP) on human dermal fibroblast cell viability. (**a**) Fibroblast cells were covered with 250 µL or 500 µL of sterile CS-βGP for 24 h. (**b**) Fibroblast cells were treated for 24 h with 0.9% NaCl previously incubated with sterile CS-βGP gel for up to 72 h to observe unwanted cytotoxic effect of components released from the gel. Fibroblast viability was measured using the CellTiter-Glo viability assay. Data are expressed as mean ± standard deviation (n = 3).

**Figure 7 pharmaceutics-14-02841-f007:**
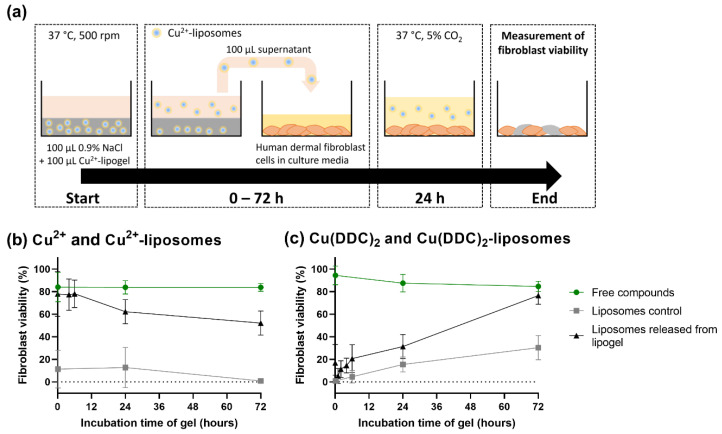
Effect on human dermal fibroblast cell viability of liposomes released from sterile chitosan with β-glycerophosphate (CS-βGP) gel over 72 h. (**a**) Experimental procedure exemplified with release of Cu^2+^-liposomes from the Cu^2+^-lipogel over 72 h. Cu^2+^-lipogel was covered with release media and incubated at 37 °C and 500 rpm. After predetermined timepoints, the supernatant was transferred onto human dermal fibroblast cells. Following 24 h incubation, fibroblast viability was measured using CellTiter-Glo viability assay. (**b**) Effect on fibroblast viability of released Cu^2+^-liposomes from Cu^2+^-lipogel over time. Controls include the effect of free Cu^2+^ incubated with sterile CS-βGP gel (corresponding to 100% released non-liposomal Cu^2+^) and of Cu^2+^-liposomes incubated with sterile CS-βGP (corresponding to 100% released Cu^2+^-liposomes). (**c**) Effect on fibroblast viability of released Cu(DDC)_2_-liposomes from Cu(DDC)_2_-lipogel over time. Controls include the effect of free Cu(DDC)_2_ incubated with sterile CS-βGP (corresponding to 100% released non-liposomal Cu(DDC)_2_) and of Cu(DDC)_2_-liposomes incubated with sterile CS-βGP (corresponding to 100% released Cu(DDC)_2_-liposomes). Data are expressed as mean ± standard deviation (n = 3).

**Figure 8 pharmaceutics-14-02841-f008:**
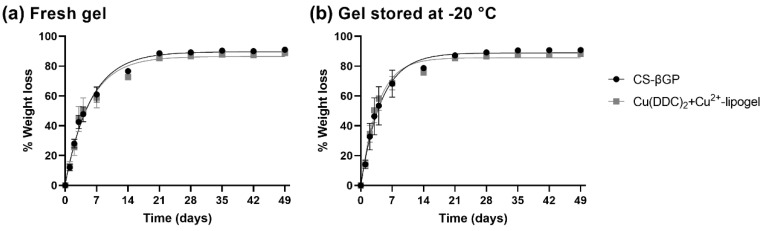
Non-linear fit of the percentage of accumulated weight loss over 49 days at 37 °C of chitosan with β-glycerophosphate (CS-βGP) gel and Cu(DDC)_2_+Cu^2+^-lipogel. Weight loss of (**a**) freshly prepared gels or (**b**) gels based on CS-βGP stored at −20 °C. Data are expressed as mean ± standard deviation (n = 3).

**Figure 9 pharmaceutics-14-02841-f009:**
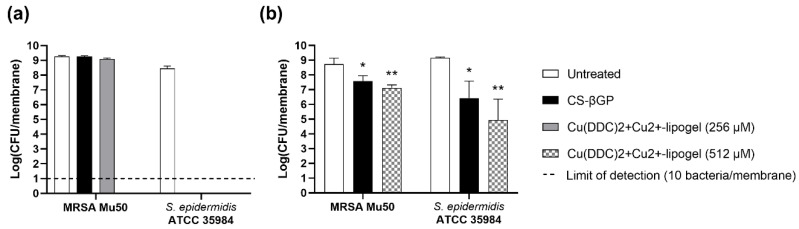
Effect of Cu(DDC)_2_+Cu^2+^-lipogel on colony-forming units (CFUs) of MRSA Mu50 and *S. epidermidis* ATCC 35984. Log(CFU/membrane) in (**a**) prevention of biofilm formation assay over 2 days and (**b**) biofilm treatment over 4 days. Missing bars represent no bacterial detection (below limit of detection; dashed line). Concentrations of Cu(DDC)_2_-liposomes + Cu^2+^-liposomes correspond to the total Cu^2+^ concentration and are based on a 1:6.2 molar ratio of Cu(DDC)_2_-liposomes to Cu^2+^-liposomes. Data are expressed as geometric mean ± standard deviation (n = 3; 1-way ANOVA with Tukey’s multiple comparison test, * *p* < 0.05, ** *p* < 0.01). CS-βGP = chitosan with β-glycerophosphate.

**Table 1 pharmaceutics-14-02841-t001:** Parameters for the lyophilization process of Cu(DDC)_2_-liposomes and/or Cu^2+^-liposomes.

Parameters	Freezing	Primary Drying	Secondary Drying	
			First Step	Second Step
Temperature (°C)	−80	−45	0	25
Pressure (mbar)	-	0.07	0.001	0.001
Time (h)	12	42	3	3

**Table 2 pharmaceutics-14-02841-t002:** Sol-gel transition temperature (°C) and time (s) of chitosan with β-glycerophosphate (CS-βGP; molar ratio 1:4.88). The sol-gel transition temperature and time were determined by rheological measurements of a temperature sweep and a time sweep mimicking an injection, respectively. During the temperature sweep, the temperature was increased from 15 to 45 °C at a rate of 2 °C/min. During the time sweep, the temperature was increased from 15 °C to 37 °C at maximum heating rate (within 75 s, mimicking an injection), and the time until sol-gel transition was either measured as total time (start point 15 °C) or time at 37 °C (start point 37 °C). ND = not determined; NR = 37 °C not reached. Data are expressed as mean ± standard deviation (n = 3–4).

CS-βGP Mix	Sterile	+ Cu^2+^-Liposomes	+Cu(DDC)_2_-Liposomes	Temperature (°C) ± SD	Total Time (s) ± SD	Time (s) at 37 °C
Freshly prepared	-	-	-	35.3 ± 3.1	ND	ND
	+	-	-	34.2 ± 2.9	68 ± 16	NR
	+	+	-	34.8 ± 0.5	70 ± 4	NR
	+	-	+	38.8 ± 1.5	330 ± 144	255
	+	+	+	33.3 ± 2.6	75 ± 14	NR
stored at −20 °C	+	-	-	39.2 ± 1.0	90 ± 25	15
	+	+	+	37.9 ± 3.3	118 ± 50	43

**Table 3 pharmaceutics-14-02841-t003:** Mean time until 50% of weight loss and rate constant based on non-linear fit of weight loss over 49 days at 37 °C of chitosan with β-glycerophosphate (CS-βGP) gel or −20 °C stored CS-βGP gel, alone or as Cu(DDC)_2_+Cu^2+^-lipogel. Data are expressed as mean with 95% confidence interval (95% CI; n = 3).

CS-βGP Gel	+ Cu(DDC)_2_-Liposomes + Cu^2+^-Liposomes	Mean Time until 50% Weight Loss [95% CI] (Days)	Rate Constant [95% CI] (1/Days)	R^2^
fresh	−	3.9 [3.5 to 4.4]	0.18 [0.16 to 0.20]	0.988
	+	3.7 [3.1 to 4.5]	0.19 [0.15 to 0.22]	0.972
−20 °C	−	3.1 [2.6 to 3.8]	0.22 [0.18 to 0.26]	0.966
	+	2.6 [2.3 to 3.1]	0.26 [0.23 to 0.30]	0.975

## Data Availability

The data presented in this study are available on request from the corresponding author.

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
