# Peer review of "A Thermosensitive, Chitosan-Based Hydrogel as Delivery System for Antibacterial Liposomes to Surgical Site Infections"

_pharmaceutics, 2022, doi:10.3390/pharmaceutics14122841_

Round 1

Reviewer 1 Report

The authors propose novel and interesting approach to attack biofilms in surgical site infections.

The work is clearly presented, the methods are justified, and discussion comprises relevant references and in-depth evaluation.

My minor comments are related to biofilm evaluation:

the choice of microorganisms can be discussed in more depth considering the challenges in mimicking biofilms

I would also extend the discussion on prevention versus eradication of biofilms

as well as challenges related to the site of infection.

Reviewer 2 Report

This is a nice, well-performed and wel-presented work. The manuscript merits publication.

Author Response

We thank Reviewer 2 for reviewing our paper. The revised manuscript was proofread for English literacy and grammatical and typographical errors were corrected.

Reviewer 3 Report

The following suggestion will be helpful to improve the further quality of the manuscript.

The present form of abstract provide generalize overview of the work, in my opinion, the finding should also be incorporated in the revised manuscript in a form of absolute results also.

2. Authors have to incorporate the formulation-related literature carried out by other researchers and highlight the current research gap. It is suggested to link the current investigation to fulfill the formulation-related research gap including the merits of your work.

3. Include an illustration highlighting the formulation step involved in the preparation of liposomes in the current investigation. 

4. It will be interesting for the reader to include the key factors which is influencing the physicochemical properties (vesicle size, lamellarity, vesicle morphology) of the developed liposome system in the current investigation.

5. It will be interesting for the reader to include a detailed discussion particularly related to the key factors which is influencing the physicochemical properties (vesicle size, lamellarity, vesicle morphology) of the developed liposome system in the current investigation.

Reviewer 4 Report

The reported work entitled " A Thermosensitive, Chitosan-Based Hydrogel as Delivery System for Antibacterial Liposomes to Surgical Site Infections” is interesting. However, the manuscript can be accepted in Pharmaceutics after taking my concerns into account, as follows.

1. Many grammatical and typographical errors must be carefully corrected.

2. There are more studies about chitosan hydrogel l formulation. What is new in this study? Please discuss the novelty of your study in the introduction clearly.

3. what is the rheological behaviour of gel?? is it gel-sol-gel?? does it shows thixotropic properties? kindly report

4. What was the pH of final formulation?

5. Perform the extrudability and spreading behaviour of your gel and compare with placebo.

Reviewer 5 Report

Dear Editor,

Please, find enclosed my comments relevant to the manuscript ID: pharmaceutics-2082153 entitled “A Thermosensitive, Chitosan-Based Hydrogel as Delivery System for Antibacterial Liposomes to Surgical Site Infections” (by Laurine Kaul, et al.).

The paper aims to the development of an injectable gel containing a liposomal formulation of  Cu(DDC)2 and Cu2+ (lipogel) with antibacterial activity.

In the following my comments.

TEM images of the prepared systems are required for a complete characterization of the prepared systems and to verify the integrity of the liposomes released from the gel.

In Figures 7(b) and 7(c) the toxicity of the control liposomes is surprising given that liposomes are generally biocompatible nanocarriers showing little cellular toxicity. Moreover this toxicity increases after 72 hours of incubation in the case of Cu+2-liposomes and decreases in the case of Cu(DDC)2-liposomes. Taking into account Cu+2 and Cu(DDC)2 show similar low toxicity and therefore the low vitality of liposomal systems is to be attributed to liposomes which have same lipid composition in the two cases, why do they show different effects on fibroblast vitality?

In line 613 the authors state "While most of the weight loss can be attributed to water escaping the gel, the increased constant rate of lipogels suggest that liposomes are released with the water from the gel structure". Proof of this claim would be desirable

Round 2

Reviewer 3 Report

The present form of revised manuscript should be consider for publication.

Reviewer 4 Report

I would like to thank the authors for adequately addressing all the comments. 

I believe that the manuscript at its current form is in good shape for publication and that the quality of presentation, as well as the scientific soundness are significantly improved. I would suggest that the authors do a final revision and correct any minor spelling errors (subscripts/superscripts were necessary, space before units, etc.) before the final submission.     

Reviewer 5 Report

Dear Editor,

the changes made to the manuscript can be considered acceptable